# A role for the *C. elegans* Argonaute protein CSR-1 in small nuclear RNA 3' processing

**Brandon M. Waddell**[1], **Cheng-Wei Wu**[1,2,3]*

**1** Department of Veterinary Biomedical Sciences, Western College of Veterinary Medicine, University of Saskatchewan, Saskatoon, Saskatchewan, Canada, **2** Toxicology Centre, University of Saskatchewan, Saskatoon, Saskatchewan, Canada, **3** Department of Biochemistry, Microbiology and Immunology, College of Medicine, University of Saskatchewan, Saskatoon, Saskatchewan, Canada

* michael.wu@usask.ca

## Abstract

The Integrator is a multi-subunit protein complex that catalyzes the maturation of snRNA transcripts via 3' cleavage, a step required for snRNA incorporation with snRNP for spliceosome biogenesis. Here we developed a GFP based *in vivo* snRNA misprocessing reporter as a readout of Integrator function and performed a genome-wide RNAi screen for Integrator regulators. We found that loss of the Argonaute encoding *csr-1* gene resulted in widespread 3' misprocessing of snRNA transcripts that is accompanied by a significant increase in alternative splicing. Loss of the *csr-1* gene down-regulates the germline expression of Integrator subunits 4 and 6 and is accompanied by a reduced protein translation efficiency of multiple Integrator catalytic and non-catalytic subunits. Through isoform and motif mutant analysis, we determined that CSR-1's effect on snRNA processing is dependent on its catalytic slicer activity but does not involve the CSR-1a isoform. Moreover, mRNA-sequencing revealed high similarity in the transcriptome profile between *csr-1* and Integrator subunit knockdown via RNAi. Together, our findings reveal CSR-1 as a new regulator of the Integrator complex and implicate a novel role of this Argonaute protein in snRNA 3' processing.

## Author summary

Small nuclear RNA molecules are an important structural component of the spliceosome, which is the cellular machinery that performs RNA splicing. Proper RNA splicing is important for ensuring proteins are accurately produced within the cell, and this process is broadly important for contributing to basic physiological processes such as development. Inside the cell, small nuclear RNA is processed by a protein complex called the Integrator, and mutations to the Integrator affect RNA splicing and can lead to neurodevelopmental defects. Here, we used the roundworm *Caenorhabditis elegans* to describe how an Argonaute encoding gene called *csr-1* is required for maintaining Integrator protein expression and that loss of *csr-1* gene expression contributes to the dysregulation of small nuclear RNA processing. These results provide new insights into our understanding of fundamental factors that regulate small nuclear RNA processing in cells, which are directly important in RNA splicing control.

**Data Availability Statement:** All datasets supporting this manuscript are publicly available and found within the article and the supporting information. All numerical data are presented in S4 Table. RNA-sequencing data generated from this

study are publicly available on the NCBI GEO depository GSE243495 (https://www.ncbi.nlm.nih.gov/geo/query/acc.cgi?acc=GSE243495).

**Funding:** This work is supported by a Natural Sciences and Engineering Research Council of Canada Discovery Grant to CWW(04486). The funders had no roles in the study design, data collection and analysis, decision to publish, or preparation of the manuscript.

## Introduction

Eukaryotic RNA splicing is catalyzed by the spliceosome that removes noncoding intron segments from pre-mRNA transcripts to produce a mature mRNA for protein translation [1]. A core component of the spliceosome is the uridylate-rich small nuclear RNA (snRNA) molecules U1, U2, U4, U5, and U6 that are incorporated within small nuclear ribonucleoprotein (snRNP) complexes that serve to facilitate splice site recognition for intron removal [1,2]. The biosynthesis of snRNA transcripts begins with transcription by RNA polymerase II to yield a pre-snRNA transcript with an extended 3' precursor [3,4]. Post transcription, the pre-snRNA transcripts are processed and cleaved by the Integrator complex at the 3' end to yield mature snRNA transcripts that are then incorporated with snRNP towards spliceosome biogenesis [5,6]. The Integrator is a metazoan conserved protein complex that is composed of at least 15 distinct subunits in humans and was discovered in 2005 as the elusive molecular machinery for snRNA 3' processing [6,7]. The Integrator catalytic module includes subunits 4, 9, and 11 that are directly involved in snRNA cleavage, while the functions of non-catalytic subunits are not well defined [8,9]. Knockdown of both catalytic and non-catalytic subunits of the Integrator leads to RNA polymerase II termination failure and 3' misprocessing of the snRNA transcripts. This results in transcriptional read-through errors that can lead to the aberrant polyadenylation of snRNA, or the synthesis of a long chimeric RNA that is composed of the snRNA transcript and its unprocessed 3' end tethered to its downstream mRNA gene [10,11].

Beyond snRNA 3' processing, recent evidence has elucidated a broader role for the Integrator in contributing to transcriptional homeostasis; these functions include the 3' processing of non-coding Piwi-interacting RNAs as well as cleavage of nascent mRNAs at RNA polymerase II paused sites to facilitate either gene transcription activation or repression [12–14]. Phenotypically, human mutations to the Integrator complex have been linked to severe neurodevelopmental syndrome and developmental ciliopathies resulting in oral-facial digital syndromes [9,15,16]. Analysis of the Cancer Genome Atlas has also revealed an increase in non-synonymous mutations to the Integrator subunits in primary tumour samples [13,17]. In model organisms, the knockdown of Integrator causes developmental arrest and results in a shortened lifespan of *C. elegans*, and depletion of Integrator in mouse results in cortical neuron migration defects leading to neurological disorders [11,18,19]. To date, while the core functions of the Integrator are well developed, regulators of the Integrator complex itself are less understood. Identifying mechanisms of Integrator regulation is of interest given its diverse influence on the transcriptome and its emergence in various human diseases.

In this study, we utilize the genetic model *C. elegans* to develop a GFP based *in vivo* snRNA misprocessing reporter as a readout for Integrator malfunction and performed a genome-wide RNAi screen to identify potential Integrator regulators. We identified a novel role for the *csr-1* (Chromsome-Segregation and RNAi deficient) gene encoding an essential Argonaute protein as a regulator of the Integrator complex in snRNA processing [20]. CSR-1 is well characterized for its core role in the protection of germline gene expression, this is achieved through a tethered interaction with target transcripts that is mediated by interfacing with small RNAs (22G-RNAs) that are antisense to the targeted gene [21,22]. More recently, a role for CSR-1 in the embryo cleavage of maternal mRNAs to facilitate clearance and removal have also been demonstrated [23]. Deletion of *csr-1* results in sterility that is accompanied by loss of P-granule formation, defects in chromosome segregation, and mis-expression of replication dependent histone proteins [20,24]. Here, we show that loss of *csr-1* results in an aberrant increase in snRNA 3' misprocessing and the alternative splicing of ~400 transcripts across the transcriptome. Mechanistically, our results show that loss of *csr-1* down-regulate expression of Integrator subunit proteins in the germline that is supported by Ribo-Seq analysis indicating a

reduced translation efficiency of Integrator subunits functioning in both the catalytic and non-catalytic domains. Together, this study provides new insights into *csr-1* as a regulator of the Integrator complex that can influence snRNA 3' processing in *C. elegans*.

## Results

### Identification of novel snRNA processing regulators

The Integrator complex serves as the principle regulator of snRNA processing in eukaryotes that catalyzes 3' post-transcriptional cleavage required for snRNA maturation [6]. Disruption of the Integrator complex has been shown to impair *C. elegans* development, and can mimic a transcriptome profile similar to cadmium exposure [11,18]. To identify novel regulators of the Integrator or snRNA processing, we developed a visual biomarker of snRNA misprocessing in *C. elegans* by adapting the strategy previously employed in the *Drosophila* S2 cells [25]. We chose to design the snRNA misprocessing reporter using the C47F8.9 transcript encoding the U2 snRNA as we previously showed that the knockdown of Integrator subunits by RNAi results in the misprocessing and increased aberrant polyadenylation of this transcript [18]. A PCR amplified genomic fragment of C47F8.9 containing the promoter, transcript, and a potential 3' motif for cleavage recognition was cloned in frame with GFP (**Fig 1A**). *C. elegans* lack a conserved 3' box sequence 9–19 nucleotides downstream of the coding region that is found in other metazoans serving as a cleavage signal for the Integrator [5,26]. As such, we cloned approximately 75 base pairs downstream of the C47F8.9 transcript which contains a potential 3' motif that is conserved across U2 snRNA transcripts. Under normal conditions, the GFP signal is absent as snRNA transcripts are cleaved by the Integrator complex resulting in the loss of GFP transcript; however, RNAi knockdown of *ints-4* encoding a catalytic subunit of the Integrator results in transcriptional read-through that strongly activates GFP expression (**Fig 1B**).

Next, we performed a genome-wide RNAi screen to identify genes that when knocked down via RNAi result in the activation of the snRNA misprocessing reporter. We screened ~19,000 genes and verified 47 genes that when silenced via RNAi result in GFP activation (**Fig 1C**). These include genes encoding additional subunits of the Integrator, those involved in nuclear membrane and transport, and regulators of siRNA processing machinery (**Fig 1D and S1 Table**). We focused on 3 genes that showed the strongest GFP activation that did not encode a known subunit of the Integrator complex, which were *csr-1*, *npp-1* (Nuclear Pore complex Protein), and *npp-6* (**Fig 1E**). To verify that the increase in GFP reporter fluorescence reflects the misprocessing of the endogenous snRNA transcript, we designed a pair of primers that measure the total and misprocessed levels of C47F8.9 (**Fig 1F**). We then knocked down *csr-1*, *npp-1*, and *npp-6* in N2 wildtype worms via RNAi and found that the knockdown of all three genes resulted in a significant increase in misprocessed levels of C47F8.9 without affecting the total transcript levels. To confirm that the processing of other snRNA transcripts was also regulated by these three genes, we employed the same qPCR strategy and found that the knockdown of *csr-1* and *npp-6* led to increased misprocessing of the U4 snRNA transcript K03B8.10 (**S1A Fig**). Overall, the results here identified several novel regulators of snRNA processing and verified a role for *csr-1* and *npp-6* as a requirement of the 3' processing of U2 and U4 snRNA transcripts.

### Isoform and domain requirements for CSR-1 in snRNA processing

We decided to focus on characterizing *csr-1* as the endogenous levels of snRNA misprocessing were the greatest compared to the *npp-6*. Since *csr-1* RNAi is predicted to knockdown both *csr-1a* and *csr-1b* isoforms given that the dsRNA targets a shared coding region, we first

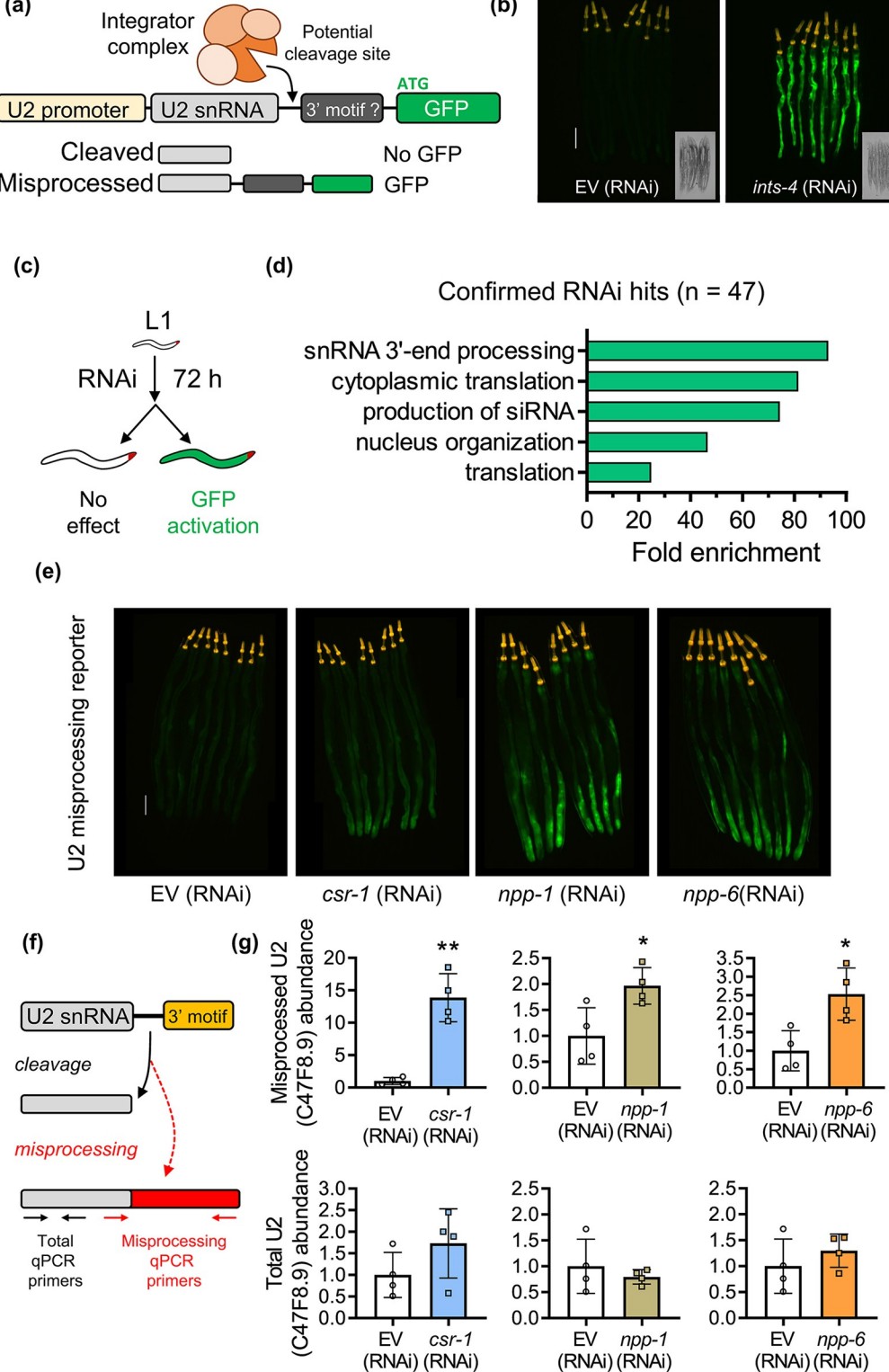

**Fig 1. Genome-wide screening of snRNA 3' processing factors. a)** Schematic of U2 snRNA misprocessing reporter. An 883bp fragment containing the U2 snRNA (C47F8.9 gene), transcript, and 3' sequence was cloned and fused to GFP and stably integrated into the *C. elegans* genome to create an *in vivo* snRNA misprocessing reporter. **b)** Representative fluorescent micrograph and brightfield image showing basal expression of the snRNA misprocessing reporter and GFP activation after Integrator disruption via *ints-4* (RNAi). The yellow signal near the pharynx represents the *myo-3p*::

tdTomato co-injection makers. **c)** Outline of the genome-wide RNAi screen to identify gene knockdowns that activate the snRNA misprocessing reporter. **d)** Enrichment analysis of 47 genes that when knocked down activate the snRNA misprocessing reporter. **e)** Representative fluorescent micrograph of the snRNA misprocessing reporter fed with EV, *csr-1*, *npp-1*, or *npp-6* RNAi. The scale bar is 100 μm. **f)** Primer design to measure total and misprocessed transcripts of the U2 snRNA. **g)** Relative levels of misprocessed and total U2 snRNA in worms fed with EV, *csr-1*, *npp-1*, *npp-6* as determined via qPCR. *P<0.05, **P<0.01 as determined by student t-test.

utilized a *csr-1a* null mutant (*cmp135*) that removes 20 bp of the coding sequence in exon 1 that is exclusively expressed by the *csr-1a* isoform (**Fig 2A**). Compared to wildtype, the *csr-1a (cmp135)* mutants do not show increased levels of misprocessed U2 or U4 snRNA transcripts (**Figs 2B** and **S1B**). We next tested the *csr-1(tm892)* mutant that contains a 400 bp deletion to both *csr-1* isoforms and found that compared to wildtype, *csr-1(tm892)* mutants showed a significantly increased level of misprocessed U2 and U4 snRNA, with the level of misprocessing comparable to those observed via *csr-1* RNAi (**Figs 2C** and **S1C**). We then examined a hypomorphic allele where *csr-1* is partially rescued in the germline and found that the levels of misprocessed U2 and U4 snRNA were also significantly elevated, albeit not to the same degree as the *csr-1(tm892)* mutant (**Fig 2D**). Together, these results suggest that the *csr-1b* isoform, but not the *csr-1a* isoform, is required for snRNA processing. However, given that *csr-1(tm892)* deletes both isoforms, we cannot rule out the possibility that snRNA misprocessing observed in this mutant is caused by the loss of both isoforms.

The CSR-1 protein encodes a Piwi domain that is catalytically active with RNA slicing activity [27]. To determine if the slicing activity of CSR-1 is required for snRNA misprocessing, we utilized two worm strains that express either a single copy insertion of wildtype (*csr-1*$^{WT}$) or slicing inactive (*csr-1*$^{SIN}$; D606A, D681A mutations) variant of *csr-1b* [28]. The two variants of the single copy *csr-1b* also contain a re-encoded region in exon 6 that renders resistance to a *csr-1* RNAi targeting 420 bp within exon 6 that is only effective against the endogenous *csr-1* gene (**Fig 2A**) [28]. Wildtype worms fed with dsRNA targeting the *csr-1* re-encoded region resulted in a significant increase in U2 and U4 snRNA misprocessing (**Figs 2E** and **S1E**). In the *csr-1*$^{WT}$ strain, RNAi against the re-encoded region did not cause an increase in U2 or U4 snRNA misprocessing, suggesting that single copy addition of RNAi resistant wildtype *csr-1b* is sufficient to compensate for the knockdown of endogenous *csr-1* (**Figs 2E** and **S1E**). In contrast, RNAi against the re-encoded region in worms expressing *csr-1*$^{SIN}$ resulted in the misprocessing of U2 and U4 snRNA to a similar extent observed in the wildtype strain (**Figs 2E** and **S1E**). This suggested that the single copy addition of a slicing inactive *csr-1b* variant does not rescue snRNA misprocessing caused by the knockdown of the endogenous *csr-1* gene (**Figs 2E** and **S1E**). Overall, the results here support the requirement of the CSR-1's enzymatic slicing activity for snRNA processing.

## *csr-1* knockdown affects transcriptome-wide snRNA processing

To investigate the role *csr-1* has on the transcriptome, we knocked down *csr-1* using RNAi and performed mRNA-sequencing of oligo(dT) enriched transcripts. An increased accumulation of snRNA transcripts beyond their 3' end is observed in the *csr-1* knocked down worms indicating transcriptional read-through as evidence for 3' misprocessing (**Fig 3A**). We found that expression for 38 snRNA transcripts can be detected in the EV control sample, and knockdown of *csr-1* resulted in a 2-fold increase in 27/38 of these transcripts (**S2A Fig**). Furthermore, 18 snRNA transcripts were detected in the *csr-1* RNAi knocked down worms that were not detected in the EV control fed worms (**S2B Fig**). Given that snRNA transcripts are only polyadenylated as a consequence of misprocessing that results in transcriptional read-through [10,18], an increased expression of snRNA abundance in oligo(dT) enriched samples also reflects an increase in snRNA

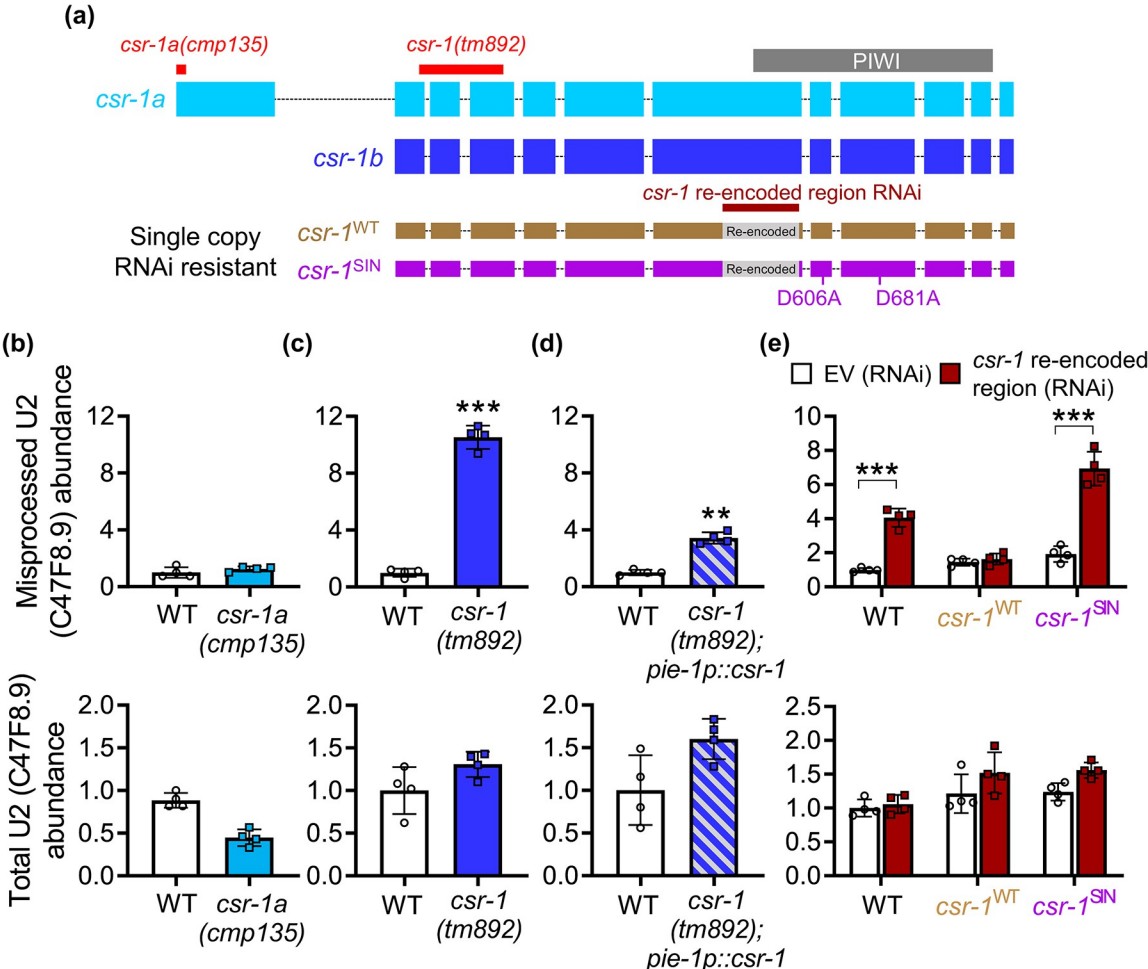

**Fig 2. Isoform analysis of *csr-1* function in snRNA processing. a)** Diagrammatic illustration of two CSR-1 isoforms, their range of nucleotide deletion in loss of function mutants, and sites of modification to RNAi resistant single copy insertion strains. Relative levels of misprocessed and total U2 snRNA in N2 wildtype (WT) and **b)** *csr-1a(cmp135)* mutant, **c)** *csr-1(tm892)* mutant, and **d)** *csr-1(tm892)* mutant with germline *csr-1* rescue. **e)** Effects of inserting a single copy of RNAi resistant wildtype *csr-1*^WT or slicing-inactive (SIN) *csr-1*^SIN on levels of misprocessed and total U2 snRNA after feeding with *csr-1* dsRNA targeting the re-encoded region. All bar graphs indicate mean ± standard error, **P<0.01 and ***P<0.001 as determined by student's t-test in **c** and **d**, and by two-way ANOVA in **e**.

misprocessing. To account for any potential developmental differences caused by *csr-1* (RNAi) in the transcriptomic data, we performed real-age prediction using transcriptome staging (RAPToR) [29]. We found that EV and *csr-1* (RNAi) samples have a predicted age of 71.82 ± 0.15 and 71.74 ± 0.16 hours, respectively, which suggest that gene expression variance between the two conditions were unlikely to be caused by potential developmental differences (**S2C Fig**). However, it should be noted that loss of *csr-1* has been shown to delay the onset of oocyte production which can lead to potential alternation of germline gene expression [30].

A recent study has shown that the transcriptional read-through effect of Integrator malfunction results in the up-regulation of genes that are located downstream of the misprocessed snRNA [11]. We compared the fold change of genes that are located directly downstream of each snRNA transcript after Integrator subunit-4 (*ints-4*) RNAi knockdown to the fold change observed for the same genes after *csr-1* RNAi knockdown and found that the expression of snRNA downstream genes is highly correlated (R = 0.626) between *ints-4* and *csr-1* knockdown (**Fig 3B**). We then performed the same analysis using RNA-seq data of the *csr-1(tm892)*

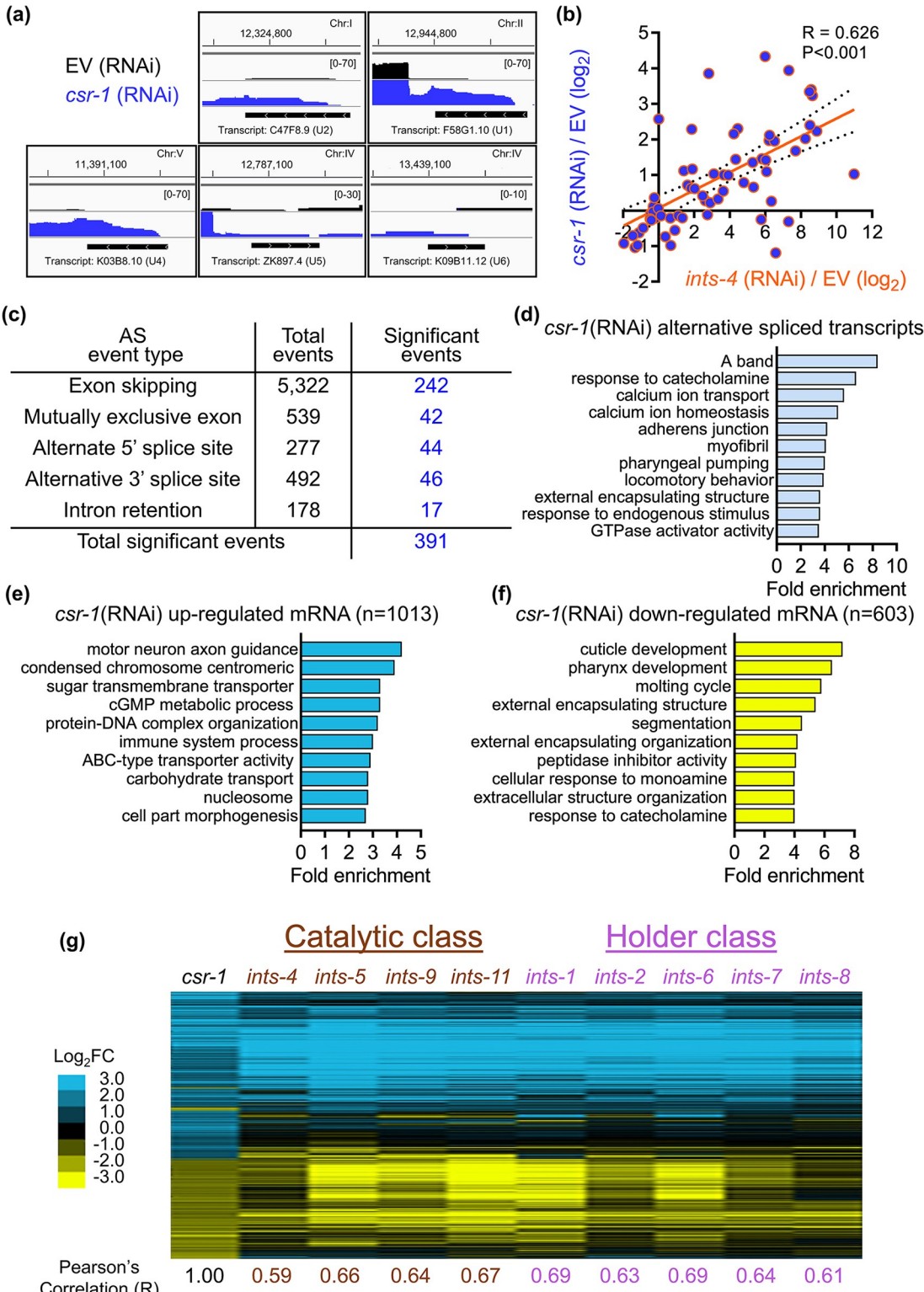

**Fig 3. The transcriptome effect of *csr-1* knockdown resembles Integrator disruption. a)** RNA sequencing reads in EV or *csr-1* (RNAi) fed worms aligned to the *C. elegans* genome in the region of different snRNA genes visualized using the Integrated Genomics Viewer. **b)** Linear regression analysis on log$_2$ fold change of snRNA downstream genes between *ints-4* (RNAi) and *csr-1* (RNAi), ***P<0.001 as determined by the F-test. N = 72 snRNA downstream genes are plotted. **c)** Number of transcripts alternatively spliced by *csr-1* (RNAi) compared to EV and **d)** Gene Ontology (GO) terms enriched by transcripts that undergo

significant alternative splicing events. Enrichment analysis of GO terms by mRNA transcripts that were **e)** up-regulated or **f)** down-regulated by >2-fold in *csr-1* (RNAi) compared to EV. **g)** Clustered heat map of $log_2$ fold change in gene expression changes caused by *csr-1* (RNAi) / EV compared to RNAi knockdown of different Integrator complex subunit genes / EV that function within the catalytic or holder class. Pearson's correlation (R) value for each gene knockdown compared to *csr-1* (RNAi) is shown below. RNA-sequencing expression data for Integrator subunit knockdown were retrieved from [11] for analysis. N = 1353 genes up or down-regulated by >2-fold after *csr-1*(RNAi) are plotted.

mutant recently reported by Singh et al. and found a similar result where expressions of snRNA downstream genes were significantly correlated between *csr-1(tm892)* mutants and *ints-4* (RNAi) (**S2D Fig**) [30]. This indicated that knockdown or loss of *csr-1* also results in similar up-regulation of snRNA downstream genes observed after *ints-4* depletion as a consequence of transcriptional read-through.

Given that *csr-1* knockdown resulted in snRNA misprocessing, we next determined the effects it has on alternative splicing. Analysis of the RNA sequencing data showed that 6,808 alternative splicing events were detected in *csr-1* knocked down worms relative to EV, with 391 of these events found to be statistically significant and composed of primarily exon skipping events (**Fig 3C**). Enrichment analysis of these 391 significant events reveals clustering to a wide range of cellular processes including calcium transport, muscle functions, and response to stimulus (**Fig 3D**). However, these enriched processes of alternatively spliced transcripts were largely distinct from the enriched cellular processes of genes that were up or down-regulated by *csr-1* RNAi (**Fig 3E and 3F**). Of the transcripts that were alternatively spliced by *csr-1* (RNAi), 60% (232/391) were differentially expressed at the mRNA levels (139 up-regulated, 93 down-regulated), which is ~2-fold higher than the global effect *csr-1* (RNAi) has on the whole transcriptome where 31% of the genes were differentially regulated (FDR<0.05, no fold change cut off). While this suggests that transcripts alternatively spliced in response to *csr-1* depletion have increased rates of differential expression, the functional significance of these alterations remains to be experimentally determined.

To further compare the transcriptome profile between *csr-1* and the Integrator, we generated a clustered heat map of expression changes for genes that were up or down-regulated by *csr-1* RNAi by > 2-fold in comparison to expression changes for these same genes after RNAi knockdown of genes encoding the catalytic and holder class of the Integrator subunit [11]. We observed a striking similarity across the 9 different Integrator subunit genes which when knocked down by RNAi exhibit a correlation value of 0.59–0.69 in the expression of genes that were differentially regulated by *csr-1* RNAi by > 2-fold (**Fig 3G**). A similar correlation in expression patterns was also observed between Integrator subunit knockdown and the previously published *csr-1(tm892)* mutant transcriptome obtained from RNA-seq and microarray studies (**S2E Fig**) [20,30]. We did not compare the gene expression changes to the auxiliary class (*ints-3*, *ints-10*, *ints-12*, and *ints-13*) as RNAi knockdown of these subunits resulted in minimal changes in gene expression compared to the EV control [11]. Given that we are comparing transcriptomic data between studies that employed different growth condition that likely introduces batch variation in the RNA samples, we next compared the differentially regulated gene profile of *csr-1* knockdown to another study that measured the effects of *ints-4* RNAi on the transcriptome [18]. Via linear regression analysis, we observed that the gene expression changes caused by *csr-1* knockdown were significantly correlated to those induced by *ints-4* RNAi from two independent studies and that the correlation values between the two studies were comparable at R = 0.49 and R = 0.59 (**S2F Fig**). Together, the RNA-sequencing data presented here show that knockdown of *csr-1* by RNAi results in the aberrant expression of snRNA transcripts, an increase in alternatively spliced transcripts, and exhibits a high similarity to the transcriptome profile induced by knockdown of the Integrator complex.

## Genetic interaction between *csr-1* and *ints-4*

Given that recent reports indicated that *csr-1b* functions primarily in the germline [31,32], we next tested for potential epistatic interactions with *ints-4* in snRNA processing via the auxin degradation system. We chose to study *ints-4* as this gene functions within the catalytic subunit of the Integrator complex and that knockdown of *ints-4* via RNAi generated the strongest developmental defect and shortened lifespan compared to the other subunits [11,18]. Via CRISPR, we endogenously tagged the C-terminal end of the *ints-4* locus with mKate2:: AID*::3xflag (referred to now as INTS-4::degron) to permit visualization of the INTS-4 protein via a fluorescent tag, and to enable auxin-mediated degradation of Integrator function by the co-expression of the plant TIR1 F-box protein (**Fig 4A**). We introduced the TIR1 protein under the control of the ubiquitously expressed *eft-3* promoter and found that exposure to 1 mM auxin starting at the L1 stage for 48 hours caused a significant increase in misprocessed U2 and U4 snRNA (**Figs 4B** and **S3**). Total levels of U2 snRNA were also elevated by 4.8-fold after INTS-4 degradation, but are considerably lower than the 111-fold increase in misprocessed U2 snRNA. We also confirmed that the metal responsive *numr-1* gene we previously found to be up-regulated in response to *ints-4* RNAi was also significantly up-regulated. To further validate that the auxin-inducible degradation of INTS-4 is phenotypically similar to previous effects as characterized by *ints-4* RNAi, we measured the effects of auxin exposure on worm development and aging. Exposure to auxin starting at the L1 stage caused a significant reduction in body length after 48 hours and a decreased lifespan of the INTS-4::degron strain in comparison to ethanol control (**Fig 4C and 4D**). These observations are consistent with the previously reported larval arrest and shortened lifespan phenotypes caused by *ints-4* RNAi [11,18].

To test for a potential epistatic relationship between *csr-1* and *ints-4*, we depleted *csr-1* via RNAi and INTS-4 via auxin to determine the effects of single or double knockdown on snRNA processing. Depletion of *csr-1* (via RNAi) or INTS-4 (via auxin) alone both significantly increased misprocessed U2 and U4 snRNA, with INTS-4 depletion expectedly showing a greater effect compared to *csr-1* depletion (**Fig 4E**). Interestingly, double knockdown of *csr-1* and INTS-4 resulted in a further increase in U2 misprocessing and U4 misprocessing relative to INTS-4 depletion alone (**Fig 4E**). This could suggest that the knockdown of *csr-1* causes snRNA misprocessing that is additive to the effects of Integrator disruption via depleting the catalytic subunit INTS-4 alone. However, it should be noted that since we did not use null alleles in this assay, potential residual functions of CSR-1 and INTS-4 may still be present in these double knockdowns.

## Loss of *csr-1* affects Integrator subunit expression

To explore the relationship between *csr-1* and the Integrator complex, we first examined 22G-RNA molecules bound to immunoprecipitated CSR-1a or CSR-1b proteins as previously reported by Charlesworth et al. [31]. We found that CSR-1b showed significant enrichment towards 22G-RNA that are antisense to 11 out of 12 transcripts encoding subunits of the Integrator complex (**Fig 5A**). In contrast, no enrichment of 22G-RNA antisense to Integrator subunits was found for CSR-1a, consistent with our data that loss of *csr-1a* does not influence snRNA processing. Interestingly, immunoprecipitated CSR-1b also shows an increase in enrichment towards 22G-RNA that are antisense to numerous snRNA transcripts, suggesting that CSR-1b may also directly interact with snRNA molecules (**S4A Fig**). It remains to be determined the functional significance of this interaction as sequence analysis of 22G-RNAs targeting snRNA shows binding to the 5' coding region instead of the potential 3' cleavage region which was observed for histone transcripts targeted by CSR-1 [24].

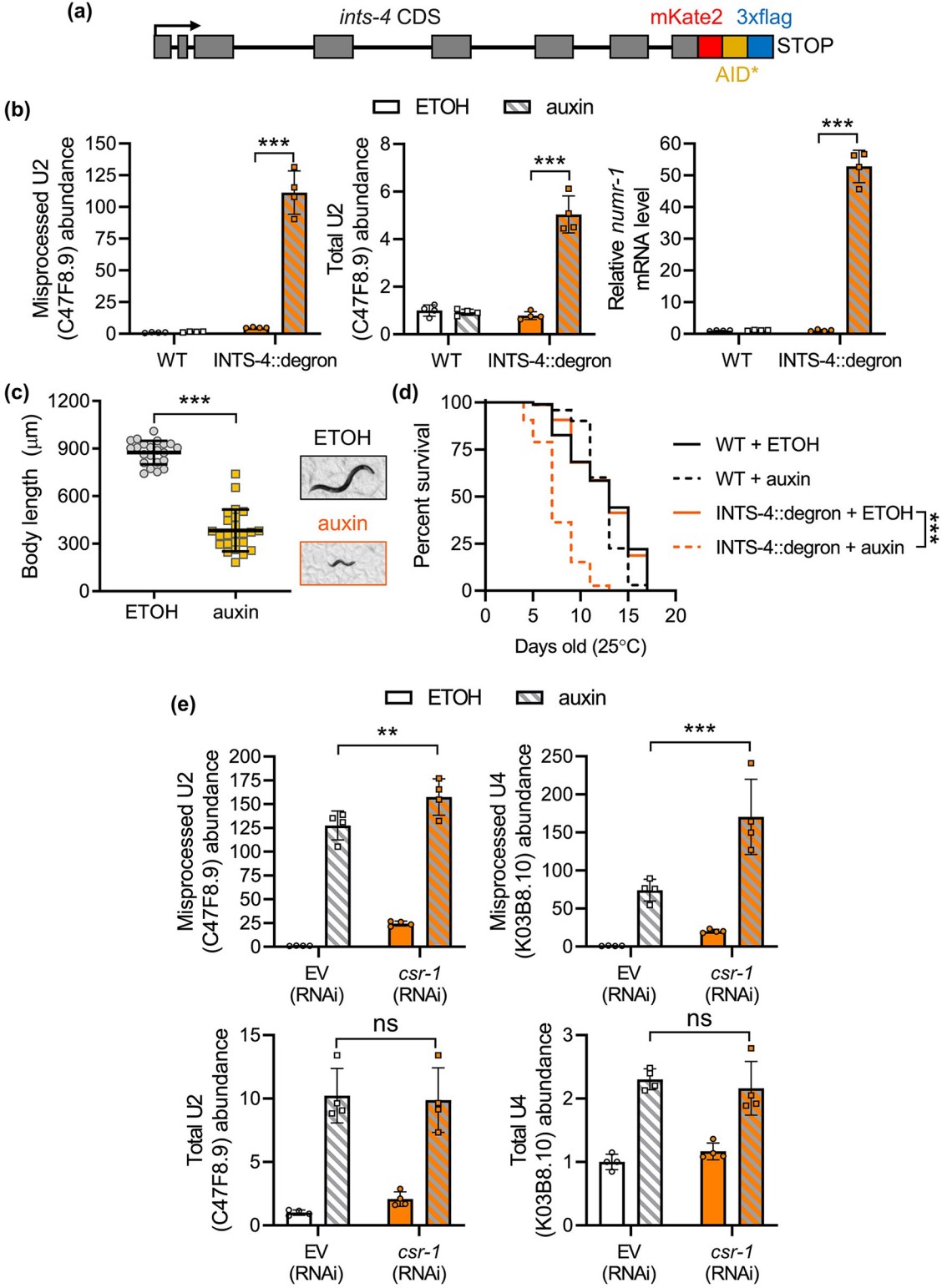

**Fig 4. Genetic interaction between *csr-1* and *ints-4* in snRNA processing. a)** Schematic illustration of the INTS-4::degron strain generated by CRISPR insertion of mKate2::AID*::3xflag in the C-terminal of the *ints-4* locus. **b)** Relative misprocessed U2, total U2, and *numr-1* transcript levels in N2 wildtype (WT) and INTS-4::degron strain expressing *eft-3p*::TIR1 treated with ethanol (ETOH) or 1 mM of auxin at the L1 stage. **c)** Effects of auxin treatment beginning at the L1 stage to INTS-4::degron; *eft-3p*::TIR1 strain on development after 48 hours compared to ETOH control. N = 135–172 worms scored for each condition. ***P<0.001 as determined

by the student's t-test. **d)** Lifespan of N2 wildtype and INTS-4::degron strain treated with ETOH or auxin from the L1 stage measured at 25°C. N = 68–139 worms scored for each condition, a summary of lifespan data is presented in S3 Table. ***P<0.001 as determined by the log-rank test. **e)** Relative levels of misprocessed and total U2 and U4 snRNA in INTS-4::degron worm strains expressing *eft-3p*::TIR1 fed with EV or *csr-1* (RNAi) at the L1 stage and treated with ETOH or 1 mM auxin for 24 hours beginning at the L4 stage. All bar graphs indicate mean ± standard error. For qPCR data in **b** and **e**,**P<0.01, ***P<0.001 as determined by two-way ANOVA.

It has been demonstrated that CSR-1 functions in a protective role of germline expression through a tethered interaction with 22G-RNAs that are antisense to the targeted transcript [21]. To explore the possibility that CSR-1 regulates the expression of Integrator subunit genes, we analyzed whole-transcriptome sequencing data between the CSR-1 KO (knockout, *tm892*) mutant and wildtype previously reported by Singh et al. [30]. In CSR-1 KO, total sRNA sequencing revealed that the majority of 22G-RNAs targeting Integrator subunits show reduced expression (**S4B Fig**), this is consistent with the report that CSR-1 is required for 22G-RNA biogenesis [30]. Analysis of mRNA-seq and GRO-seq data of CSR-1 KO mutant revealed a lack of consistent patterns in the mRNA levels and nascent transcription of genes encoding Integrator subunits (**Fig 5B and 5C**). The minimal changes to steady-state mRNA levels of Integrator subunit genes were also observed in our *csr-1* (RNAi) mRNA sequencing results and the previously published *csr-1(tm892)* microarray data (**S4C Fig**) [20]. In the Ribo-seq data, however, a decrease in gene abundance at active ribosomes was observed for 8 out of 12 Integrator subunits in the CSR-1 KO mutant, suggesting that loss of *csr-1* decreases the translation efficiency of multiple Integrator subunit proteins (**Fig 5D**).

As we showed that the catalytic activity of CSR-1 was required for snRNA processing, we next examined the sequencing dataset of the CSR-1 catalytic dead mutant (termed CSR-1 ADH) reported from the same study [30]. We found that for all three sequencing methods, the effects between CSR-1 KO and CSR-1 (ADH) mutants on Integrator subunit expressions were highly consistent (**Fig 5B–5D**), as supported by a correlation value of 0.82, 0.85, and 0.81 between CSR-1 KO and CSR-1 (ADH) for RNA-seq, Gro-seq, and Ribo-seq, respectively (**S4D–S4F Fig**). Together, these data show that loss of CSR-1 expression or its catalytic activity has a robust influence on the translation of select Integrator subunit proteins.

A role for the Integrator in piRNA processing was recently demonstrated which showed that RNAi knockdown of *ints-11* caused a global decrease in piRNA expression [12]. Given that our data suggest that *csr-1* is required for Integrator gene expression, we examined the effects of CSR-1 KO on piRNA expression. In contrast, we observed that CSR-1 KO mutants exhibit an overall increase in the global expression of mature piRNA transcripts relative to the wildtype (**S4G Fig**). The requirement of other Integrator subunits beyond *ints-11* in piRNA processing has not yet been explored, and curiously, CSR-1 KO mutants show an increase in *ints-11* abundance in Ribo-seq compared to a general trend of decrease in translation efficiency of other subunits (**Fig 5D**). As such, it remains to be determined whether the global increase in piRNA expression in CSR-1 KO mutants is influenced by its effects on *ints-11* expression, or via other potential pleiotropic functions that CSR-1 may exert in the germline that can affect piRNA expression.

To confirm the loss of *csr-1* affects Integrator protein expression, we introduced the INTS-4::degron strain that is also tagged to the mKate2 fluorescent protein into the *csr-1(tm892)* mutant background. Fluorescent microscopy indicates that INTS-4 is broadly expressed in many tissues of the worm, and is most obvious in the anterior region, germline, and hypodermis tissues (**S5A Fig**). Treatment with auxin abolishes the mKate2 signal across all tissues, indicating that the fluorescence is associated with INTS-4 (**S5A Fig**). Next, we compared the expression of INTS-4 in the wildtype and *csr-1(tm892)* mutant worm across the 3 regions

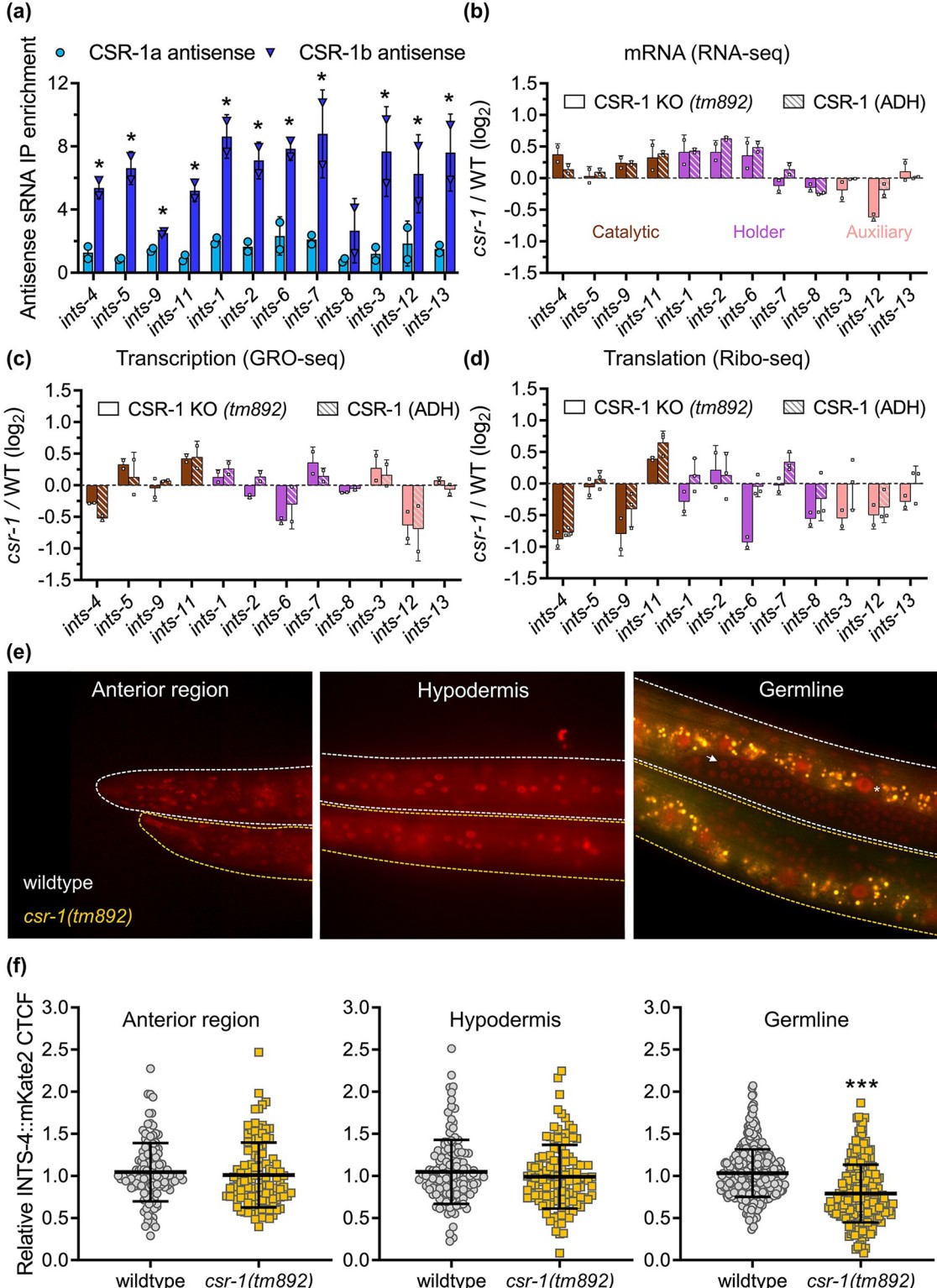

**Fig 5. Loss of *csr-1* alters Integrator subunit expression. a)** Enrichment of antisense 22G-RNAs complementary to genes encoding Integrator subunits bound to CSR-1a or CSR-1b as determined by small RNA-sequencing data (N = 2 biological replicates) obtained from [31]. Log$_2$ fold change of CSR-1 KO *(tm892)* or CSR-1 (ADH) compared to N2 wildtype (WT) for **b)** mRNA expression as determined via RNA-seq, **c)** nascent transcription as determined by GRO-seq, and **d)** mRNA translation efficiency as determined by Ribo-seq (N = 2 biological replicates). Data from **c-d** were analyzed from [30]. All bar graphs indicate

mean ± standard error, **e)** Representative fluorescent micrograph and **f)** corrected total cell fluorescence (CTCF) quantification of INTS-4::mKate2 expression in the anterior, hypodermis region, and germline in the wildtype or *csr-1(tm892)* mutant background. Two independent imaging trials were performed with N = 112–128 nuclei quantified for the anterior region, N = 120–132 nuclei quantified for the hypodermis, and N = 254–454 nuclei quantified for the germline. ***P<0.05 as determined by student's t-test. The germline composite image was created by merging images taken from the RFP and GFP filters to illustrate the non-specific intestinal gut auto-fluorescent signal represented in yellow. INTS-4 signal in the germline is marked by a white arrow and in the intestine is marked by an asterisk.

where mKate2 can be reliably quantified. INTS-4 signal is also observed in the intestine but was not quantified due to the high levels of auto-fluorescence from gut granules observed in the intestinal tract. We found that *csr-1(tm892)* mutants showed normal expression of INTS-4 in the anterior region and hypodermal cell nuclei, but demonstrated a significant reduction in fluorescence within the germline (**Fig 5E and 5F**). To rule out the potential contribution of gut auto-fluorescence to germline INTS-4 quantification due to the proximity of these tissues, we dissected the germline from wildtype and *csr-1(tm892)* mutant and measured INTS-4 fluorescence. In the dissected germlines, we observed a similar decrease of INTS-4 fluorescence in the *csr-1(tm892)* mutant compared to the wildtype and the relative fold change was consistent with the *in vivo* quantification (**S5B Fig**). A caveat to this analysis is that we did not use Western blot to measure protein expression which may provide a more robust quantification of the precise INTS-4 expression difference between wildtype and the *csr-1(tm892)* mutant.

To explore if loss of *csr-1* affects the expression of other Integrator subunits, we chose to measure the expression of INTS-6 given that *ints-6* also showed a decrease in Ribo-seq abundance in the *csr-1(tm892)* mutant (**Fig 4D**). Using a strain of worm expressing INTS-6::eGFP under its native promoter, we observed a significant decrease of *in vivo* INTS-6::eGFP fluorescence in the germline of the *csr-1(tm892)* mutant compared to the wildtype (**Fig 6A**). To determine if the additive effects of snRNA misprocessing observed when INTS-4 and *csr-1* are simultaneously depleted may be caused by the decrease in translation efficiency to multiple Integrator subunits in the *csr-1* mutant, we knocked down *ints-6* via RNAi and depleted INTS-4 via auxin. We found that co-depletion of Integrator subunits 4 and 6 exacerbated the degree of U2 snRNA misprocessing, total U2 expression, and *numr-1* activation compared to depletion of the individual subunit alone (**Fig 6B**). The effect of multiple Integrator subunit knockdown on expression changes is of synergistic nature, and may be related to the structural dependency of Integrator subunits on each other where disruption of a single Integrator subunit may compromise the activity and function of the associating subunits. For example, it is demonstrated that INTS-4 serves as an anchor for the catalytic module that contains subunits 9 and 11 while INTS-6 interacts and stabilizes with subunits 2, 5, and 8 to form the Integrator backbone core [33]. Together, these data suggest that *csr-1* contributes to snRNA processing by regulating the expression of multiple Integrator subunits.

Overall, the results in this study show that CSR-1b is bound to 22G-RNA antisense to Integrator subunit genes, and that loss of *csr-1* negatively impacts the translation efficiency of Integrator subunit genes in active ribosomes that can disrupt the post-transcriptional cleavage of snRNA.

## Discussion

The Integrator is a metazoan specific multi-protein complex that was initially discovered as the elusive termination machinery that facilitates the 3' processing of U-rich snRNAs [6]. Human mutations to the Integrator complex are characterized by increased levels of misprocessed U-rich snRNA transcripts that are accompanied by disruptions to gene expression and RNA processing [15]. While recent studies have expanded on the core functions of the

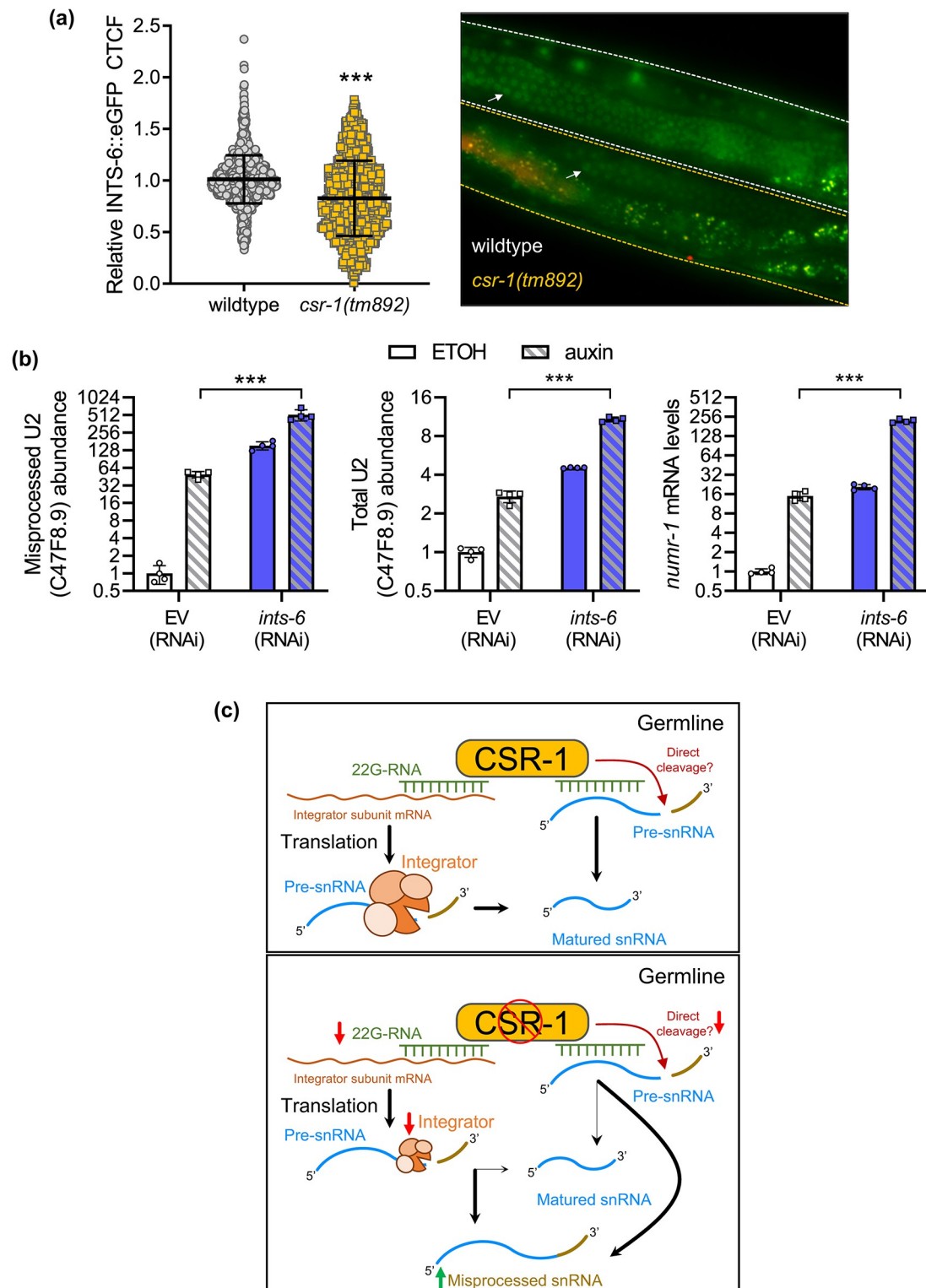

**Fig 6. Co-contribution of Integrator subunits 4 and 6 to snRNA processing. a)** Representative fluorescent micrograph (GFP and RFP merged) and CTCF quantification of INTS-6::eGFP expression in the germline of wildtype or *csr-1(tm892)* mutant worms. The bar graph indicates mean ± standard error with N = 1743 and N = 979 nuclei quantified for wildtype and *csr-1 (tm892)* respectively, ***P<0.05 as determined by student's t-test. INTS-6 signal in the germline is marked by a white arrow. **b)** Relative levels of misprocessed U2 snRNA, total U2 snRNA, and *numr-1* in INTS-4::degron worm strains expressing *eft-3p*::TIR1

fed with EV or *ints-6* (RNAi) at the L1 stage and treated with ETOH or 1 mM auxin for 24 hours beginning at the L4 stage. All bar graphs indicate mean ± standard error, ***P<0.001 as determined by two-way ANOVA. **c)** Proposed mechanism of snRNA processing regulation by CSR-1. CSR-1 in the germline binds to 22G-RNA targeting transcripts encoding subunits of the Integrator complex and this interaction supports the translation of Integrator proteins required for snRNA processing. CSR-1 also binds to 22G-RNA targeting snRNA transcripts suggesting a possible alternative mechanism via direct cleavage. Loss of CSR-1 function or catalytic activity decreases the abundance of 22G-RNA targeting the Integrator and reduces the translation efficiency of Integrator subunits leading to increased accumulation of misprocessed snRNA transcripts.

Integrator beyond snRNA processing including cleavage of nascent mRNAs during RNA polymerase pause-release [14], additional factors that regulate the Integrator complex or influence snRNA 3' processing remain underexplored. In this study through the use of an *in vivo* snRNA misprocessing reporter in the *C. elegans* system, we identified several genes, including the Argonaute encoding *csr-1*, that when knocked down result in the misprocessing of snRNA transcripts. We propose that *csr-1* is required for the germline expression of Integrator subunit proteins, and that loss of *csr-1* contributes to snRNA misprocessing by altering the abundance of Integrator complex subunits (**Fig 6C**). Additionally, given that CSR-1 also binds to 22G-RNA targeting snRNA transcripts, it is possible that CSR-1 can directly cleave snRNA molecules given its catalytic slicing activity, a function that has been proposed for the 3' processing of histone transcripts in *C. elegans* [24,27,28,30].

## CSR-1 isoforms and slicing activity in snRNA 3' processing

The CSR-1 Argonaute protein is composed of two isoforms, CSR-1a and CSR-b, which have been recently shown to exhibit distinct expression patterns and interact with diverse 22G-RNAs to regulate unique downstream targets [31,32]. CSR-1a expression is more readily detected during adulthood in the spermatogenesis region of the germline and somatic tissues including the intestine, whereas CSR-1b is constitutively detected throughout all developmental stages but strictly expressed in the germline [31,32]. While both CSR-1 isoforms are involved in fertility regulation, loss of *csr-1b* results in complete sterility and loss of *csr-1a* contributes to loss of sperm-based fertility in a transgenerational manner [31,32]. Our mutant analysis shows that snRNA transcripts are not affected in the *csr-1a* mutant but are misprocessed in the *csr-1(tm892)* mutant that contains deletion to both isoforms. This suggests that loss of function to *csr-1b*, but not *csr-1a*, impairs snRNA processing. However, given that a mutant with loss of function to only *csr-1b* is not possible, it is also conceivable that the snRNA misprocessing observed in *csr-1(tm892)* mutant is caused by the loss of both isoforms. An isoform specific role of CSR-1 is supported by the co-IP experiments showing that only CSR-1b, but not CSR-1a, binds to endo-siRNAs that are antisense to Integrator subunit genes [31]. Furthermore, we found that the introduction of an RNAi resistant slicing-inactive (SIN) variant of *csr-1b* was not able to rescue snRNA misprocessing induced by RNAi knockdown of endogenous *csr-1*. Given that the slicing activity has been reported to be essential for CSR-1 in catalyzing the biogenesis of 22G-RNA antisense to its germline targets, this data proposes a model where CSR-1b contributes to snRNA processing by targeting Integrator subunit transcripts via endo-siRNA interaction to governor germline expression [28,30]. A role for CSR-1 as a requirement in maintaining the expression of Integrator subunits is supported by the evidence that genes that are differentially regulated after *csr-1* knockdown are also similarly affected upon Integrator subunit knockdown (**Figs 3G and S2E**). In the future, it will be of interest to be determined whether these gene expression changes are linked to snRNA misprocessing which affects downstream RNA splicing, or are a consequence of disruption to Integrator's role in nascent mRNA cleavage during RNA polymerase pause-release that affects transcriptional efficiency [34].

## CSR-1 in histone and snRNA processing

Akin to snRNA transcripts, replication dependent histone mRNA is also unique in that they are not polyadenylated after transcription, but rather undergoes 3' post-transcriptional cleavage for maturation [5,10]. While the 3' processing of histone genes is primarily carried out by the stem-loop binding protein (SLBP) complex, depletion of the Integrator has also been shown to cause aberrant histone polyadenylation, suggesting that the Integrator controls 3' termination of diverse classes of transcripts beyond snRNA [10,18,35]. In this study, we identified a role for *csr-1* in snRNA processing, this is intriguing given that *csr-1* has also been previously demonstrated to process 3' cleavage of histone transcripts in *C. elegans* [24]. In the histone mechanism, it is proposed that CSR-1 binds to antisense endo-siRNA that are complementary to sequences adjacent to the stem-loop region to facilitate 3' processing [24]. Direct cleavage activity of CSR-1 on histone transcripts has not been demonstrated, but it was shown that CSR-1 binds to histone mRNA and knockdown of *csr-1* results in an increased abundance of misprocessed histones that is accompanied by a decrease in histone protein expression [24]. While CSR-1 also binds to endo-siRNA that are antisense to snRNA transcripts, the majority of these 22G-RNA are complementary to the 5' coding region of snRNA transcripts instead of the potential 3' cleavage region downstream of the coding sequence, and the significance of this interaction has not been elucidated [32]. Rather, our data suggests that CSR-1 may indirectly influence snRNA processing by regulating the abundance of Integrator proteins; however, we do not rule out the possibility that CSR-1 may also facilitate direct post-transcriptional 3' processing in a similar manner that is proposed for histone transcripts (**Fig 6C**). We show in this study that loss of *csr-1* influences INTS-4 and INTS-6 expression in the germline, and this is consistent with the known role of CSR-1 in licensing the expression of germline-expressed protein encoding genes in *C. elegans* [21]. While our study only directly analyzed INTS-4 and INTS-6 expression, Ribo-seq analysis has revealed that loss of *csr-1* results in the decreased translation efficiency of multiple Integrator subunits that function within the catalytic (*ints-4*, *ints-9*), holder (*ints-1*, *ints-6*, *ints-8*), and auxiliary (*ints-3*, *ints-12*, *ints-13*) modules [30]. This decrease in expression of multiple Integrator subunits could also explain why simultaneous knockdown of *csr-1* and INTS-4 caused an additive effect on snRNA misprocessing compared to depletion of INTS-4 alone (**Fig 4E**), as knockdown of *csr-1* decreased the translation of multiple Integrator subunits, each of which may contribute to overlapping or distinct cleavage activity on snRNA 3' processing [11]. This is supported by our data showing that co-depletion of Integrator subunits 4 and 6 exacerbates snRNA misprocessing compared to single subunit disruption (**Fig 6B**). Overall, these evidence illustrates the broad regulatory role of CSR-1 Argonaute protein in the post-transcriptional processing of histone and snRNA transcript maturation, potentially via a direct and indirect mechanism respectively.

## snRNA regulators beyond CSR-1

Our genome-wide RNAi screen identified a total of 47 genes that are required for snRNA processing as determined by the *in vivo* snRNA misprocessing reporter, 43 of which were not direct subunits of the Integrator and function in processes such as nuclear organization and siRNA biogenesis. Next to *csr-1*, we verified that the knockdown of *npp-1* and *npp-6* encoding nuclear pore protein that facilitates nucleocytoplasmic transport resulted in the misprocessing of endogenous U2 and U4 snRNA transcripts (**Fig 1G**) [36]. A direct role for nuclear pore proteins in snRNA processing has yet to be demonstrated, however, it has been shown that snRNA transcripts are briefly exported from the nucleus to the cytoplasm for modification before re-entry into the nucleus for incorporation with snRNP for spliceosome biogenesis

[37]. It is possible that the depletion of *npp* genes interferes with this step in nucleocytoplasmic shuttling and disrupts the post-transcriptional maturation of snRNA transcripts. Alternatively, the knockdown of *C. elegans npp* genes has also been shown to disrupt the germline P granule formation, which is the primary subcellular location where CSR-1 is enriched to facilitate 22G-RNA loading for antisense gene targeting [20,38]. As such, the knockdown of *npp* may contribute to snRNA misprocessing indirectly by interfering with CSR-1 function through the disruption of P granule integrity and formation.

Our RNAi screen also identified genes that encode various components of the RNAi machinery involved in the 26G-RNA pathway including *mut-16* (<u>MUT</u>ator), *dcr-1* (<u>Di</u><u>C</u>er <u>R</u>elated), *rde-4* (<u>R</u>NAi <u>DE</u>fective), *alg-4* (<u>A</u>rgonaute <u>L</u>ike <u>G</u>ene) [39,40]. While we did not verify the degree of endogenous snRNA misprocessing caused by knockdown of these genes via qPCR, the activation of the misprocessing reporter may suggest that the defect in endogenous siRNA pathways is linked to snRNA transcript misprocessing. As such, it appears that additional and potentially more complex mechanisms of snRNA regulation via the 26G-RNA pathway beyond the CSR-1 mediated 22G-RNA mechanism may exist that require future investigations.

## Conclusions

Overall, we demonstrate in this study a positive role for the *csr-1* gene encoding the only essential Argonaute protein in *C. elegans* as a regulator of snRNA processing, through a mechanism where *csr-1* is required for the translation and expression of Integrator subunit genes within the germline. Beyond *csr-1*, the genome-wide RNAi screen presented in this study has also identified several yet to be characterized regulators of snRNA processing including those encoding nuclear protein complex as well as members of the endogenous siRNA pathway. Given the recent expansion of a wide-ranging role for the Integrator complex in gene expression control beyond snRNA processing, and its emerging implication in human diseases [9,15,16], it will ultimately be of interest to determine whether these novel regulators may also influence snRNA independent functions of the Integrator in contributing to transcriptome stability.

## Materials & methods

### *C. elegans* strains

All *C. elegans* strains were cultured at 20°C using standard methods unless noted otherwise [41]. The following strains were used: N2 Bristol wildtype, MWU3 *cwwIs1[C47F8.9p:: C47F8.9::GFP; myo-2p::tdTomato]*, USC1258 *csr-1a(cmp135)*, WM182 *csr-1(tm892) IV; nT1 [unc-?(n754) let-?](IV;V)*, WM194 *csr-1(tm892) IV; neIs20 [pie-1::GFP::csr-1 + unc-119(+)]*, OD923 *ltSi240[csr-1p::csr-1(re-encoded) + Cbr-unc-119(+)] II; unc-119(ed3) III*, OD925 *ltSi242 [csr-1p::csr-1(re-encoded; D606A, D681A: isoform b numbering) + Cbr-unc-119(+)] II; unc-119 (ed3) III*, MWU193 *cwwSi1[ints-4::mKATE2::AID\*::3xFLAG]; wrdSi23 [eft-3p::TIR1::F2A:: mTagBFP2::AID\*::NLS::tbb-2 3'UTR] (I:-5.32)*, MWU200 *cwwSi1[ints-4::mKATE2::AID\*::3x-FLAG]; wrdSi23[eft-3p::TIR1::F2A::mTagBFP2::AID\*::NLS::tbb-2 3'UTR] (I:-5.32); csr-1 (tm892) IV/nT1 [unc-?(n754) let-?] (IV;V)*, EG9882 *F53A2.9(oxTi1127 [mex-5p::Cas9(+smu-2 introns)::tbb-2 3'UTR + hsp-16.41p::Cre::tbb-2 3'UTR + myo-2p::2xNLS::cyOFP::let-858 3'UTR + lox2272] III.)*, JCP341 *jcpSi10[ints-6p::ints-6::3xFLAG::eGFP::ints-6 3'UTR + unc-119(+)] II*, MWU233 *jcpSi10; csr-1(tm892) IV; nT1[unc-?(n754) let-?](IV;V)*. The MWU3 strain was generated by cloning an 883 bp fragment containing the U2 snRNA (C47F8.9 gene) promoter, transcript, and 3' downstream sequence that was then fused in frame to the GFP fluorescent protein [42]. This construct was microinjected into *C. elegans* and stably integrated into the

genome via U.V. irradiation. The stably integrated snRNA misprocessing strain was outcrossed 4 times before use.

## Genome-wide RNAi screen and RNAi experiments

RNAi screen was performed using a protocol previously described in detail [43]. Briefly, synchronized L1 MWU3 larvae obtained from hypochlorite treatment were grown in liquid nematode growth media (NGM) and fed with dsRNA producing *HT115(DE3)* bacteria for 3 days, followed by manual screening for snRNA misprocessing reporter GFP activation using an Olympus SZX61 stereomicroscope. The MRC genomic RNAi feeding library (Geneservice, Cambridge, UK) and the ORFeome RNAi feeding library (Open Biosystems, Huntsville, AL) were used totaling approximately 19,000 clones screened. Clones that activated the snRNA misprocessing reporter from the primary screen were rescreened three additional times using solid NGM agar plates for confirmation. NGM agar RNAi plates were prepared with 50 µg mL$^{-1}$ carbenicillin and 100 µg mL$^{-1}$ of isopropyl β-D-thiogalactopyranoside (IPTG) and seeded with *HT115(DE3) E. coli* expressing the corresponding target dsRNA clone or expressing the pPD129.36(L4440) plasmid that serve as the empty vector (EV) control.

## Microscopy and fluorescent analysis

For the snRNA misprocessing reporter, worms were synchronized at the L1 stage and fed with the corresponding RNAi for 72 hours followed by imaging using a Zeiss Axioskop 50 microscope fitted with a Retiga R3 camera. Worms were mounted on a glass slide containing a 2% agar pad and immobilized in a 2% sodium azide solution dissolved in the M9 buffer. To image INTS-4 tagged with mKate2 and INTS-6 tagged with eGFP *in vivo*, the worms were synchronized and grown to the L4 stage and immobilized with 0.65% sodium azide on microscope slides containing a 2% agarose pad and imaged using the Delta Vision (GE) deconvolution system. To image INTS-4::mKate2 in the dissected germline, L4 stage worms were paralyzed with M9 buffer containing 10 mM of levamisole hydrochloride in the cavity of a concave microscope slide followed by dissection with two 25 gauge needles just below the pharynx to extrude the germline follow by imaging with the Delta Vision (GE) deconvolution system. Fluorescence intensity was determined by the measure function in ImageJ and used to calculate CTCF as defined by [integrated density–(area of selection *x* mean background fluorescence)]; each value was divided by the median value of the wildtype control to determine relative CTCF. Background fluorescence was determined for each image by measuring the signal intensity in an area of the image where fluorescence was absent. Grayscale images taken were colourised in ImageJ using the merge channel function.

## RNA extraction, qpcr, and RNA-sequencing

RNA was extracted by using the Purelink RNA mini kit (ThermoFisher, 12183020) with worm lysis accomplished with a QSonica Q55 sonicator. For each condition, N = 4 biological replicates were prepared with each replicate containing approximately 500–1000 worms. For RNA extraction of the *csr-1(tm892)* strain, 100 non-Unc worms were manually picked for each replicate from a synchronized population for RNA extraction. RNA extracted for qPCR analysis was first treated with DNAseI (ThermoFisher, EN0521) followed by cDNA synthesis with the Invitrogen Mutiscribe reverse transcriptase system (ThermoFisher, 4311235) using an Applied Biosystems ProFlex Thermocycler. A QuantStudio 3 system was used to perform qPCR with the PowerUp SYBR Green Master Mix (ThermoFisher, A25741). Relative gene expression was normalized to the housekeeping gene *cdc-42*. Primers used for qPCR are listed in **S2 Table**.

For whole-transcriptome RNA sequencing, RNA was extracted using the same methods described above with the exception that each sample contained ~3,000 worms and that 3 biological replicates were prepared and sequenced for each condition. The RNA samples were sent to Novogene (Sacramento, CA) on dry ice followed by cDNA library construction with the oligo(dT) enrichment method for mRNA sequencing. Sequence annotation and data analysis were performed by Novogene. Mapping of sequence reads to the WBcel235 genome was performed with HISAT2 (v2.0.5), gene quantifications were performed with FeatureCounts (V1.5.0-p3), DESeq2 (v1.20.0) was used to analyze differentially expressed genes, and alternative splicing was analyzed by rMATS v3.2.5 [44–46]. For age estimation, normalized counts of genes from EV and *csr-1* (RNAi) fed worms were analyzed using RAPToR R package (V1.2) with the wormRef database (V0.5) of Cel_YA_1 as a reference [29].

### CRISPR/Cas9 genome editing and auxin experiments

To insert mKate2::AID*::3xflag in the C-terminus of the *ints-4* locus, we followed the SEC-based protocol described in [47,48]. Briefly, 5' and 3' homology arm flanking the C-terminal insertion site was amplified from N2 wildtype genomic DNA using the Q5 High-Fidelity DNA Polymerase (NEB, M0491L), followed by HiFi DNA assembly (NEB, E5520S) with the pJW1589 plasmid that was digested with AvrII and SpeI to assembly the repair template. The *ints-4* sgRNA was inserted into plasmid pDD162 via the Q5 Site-Directed Mutagenesis Kit (NEB, E0554S). The sgRNA plasmid and repair template were Sanger sequenced to confirm correct construct assembly followed by microinjection into EG9882 strain with integrated Cas9 activity (10 ng μl$^{-1}$ repair template, 50 ng μl$^{-1}$ sgRNA plasmid) [49]. Primers used for sgRNA insertion into pDD162 and for generation of 5' and 3' homology arms are listed in **S2 Table**. Worms homozygous for the roller phenotype were outcrossed with N2 wildtype to remove the integrated Cas9 background followed by heat shock to remove the SEC cassette. Three additional rounds of outcross with the N2 wildtype background were performed to remove any potential non-specific edits. Worms expressing the mKate2::AID*::3xflag insertion to the *ints-4* locus were crossed with JDW225 to introduce the TIR1 protein expressed by the ubiquitous *eft-3* promoter.

For auxin experiments, a 400 mM stock of indole-3-acetic acid 98% (IAA, referred to as auxin) (MilliporeSigma, I3750-25G-A) dissolved in 100% ethanol (ETOH) was used to produce NGM agar plates with a final concentration of 1 mM of auxin. NGM agar plate containing 0.25% ETOH only was used as the corresponding control. Worms were exposed to auxin either at the L1 stage or the L4 stage as described in the figure caption.

### Developmental and lifespan analysis

For the developmental assay, synchronized L1 MWU193 worms were grown on NGM agar plates containing 0.25% ETOH or 1 mM auxin and imaged with an Olympus SZX61 stereomicroscope fitted with a Retiga R3 camera after 48 hours to determine the body length. For the lifespan assay, synchronized L1 wildtype or MWU193 worms were grown on NGM agar plates containing 0.25% ETOH or 1 mM auxin at 25°C. The first day of adulthood is marked as 1 day old and adult worms were separated from their progeny via manual picking. Worms were scored every 1–2 days for death via gentle prodding with a sterilized metal pick. Worms were considered dead if they did not respond to the prodding and were censored if they exhibited protruding vulva or gonads. Lifespan assays were performed at 25°C as we observed a high rate of censorship due to vulva protrusion when the lifespan assay was initially performed at 20°C in the MWU193 strain.

## Statistical analyses

The GraphPad Prism software (V7.04) was used to generate graphical data and perform statistical analysis. Student's t-test was used when comparing two groups, one-way ANOVA with Dunnett's test was used for comparison of more than two groups, two-way ANOVA with Holm-Sidak's multiple comparison was used for assessment of two factors with multiple groups, the F-test was used to calculate statistical significance to linear regression data, lifespan data were analyzed using the log-rank test via OASIS2 (https://sbi.postech.ac.kr/oasis2) [50], and the false discovery rate (FDR) correction was applied to determine statistical significance of RNA-sequencing data.

## Supporting information

**S1 Fig. Regulation of the U4 snRNA transcript by genes identified from the RNAi screen. a)** Relative levels of misprocessed and total U4 snRNA in worms fed with EV, *csr-1*, *npp-1*, *npp-6* RNAi as determined via qPCR. Relative levels of misprocessed and total U4 snRNA in N2 wildtype (WT) and **b)** *csr-1a(cmp135)* mutant, **c)** *csr-1(tm892)* mutant, and **d)** *csr-1(tm892)* mutant with germline *csr-1* rescue. **e)** Effects of inserting a single copy of RNAi resistant wild-type *csr-1*$^{WT}$ or slicing-inactive (SIN) *csr-1*$^{SIN}$ on levels of misprocessed and total U4 snRNA after feeding with *csr-1* dsRNA targeting the re-encoded region. All bar graphs indicate mean ± standard error, *P<0.05 **P<0.01, ***P<0.001 as determined by student t-test in **a** to **d**, and by two-way ANOVA in **e**.
(TIF)

**S2 Fig. Depletion of *csr-1* leads to a widespread increase in snRNA abundance. a)** Normalized count of snRNA transcripts in poly-A-tailed mRNA sequenced RNA samples in worms fed with EV as compared to *csr-1* RNAi. The dotted line indicates 2 fold increase in expression. **b)** Normalized count of snRNA transcripts in worms fed with *csr-1* RNAi that was not detected in worms fed with EV. **c)** Age estimation of EV and *csr-1* RNAi fed worm RNA samples based on the expression profile of 25,902 genes detected via RNA-sequencing as determined via RAPToR [29]. **d)** Linear regression analysis on log$_2$ fold change of snRNA downstream genes between *ints-4* (RNAi) and *csr-1(tm892)* using data from [30], N = 71 snRNA downstream genes are plotted. **e)** Clustered heat map of log$_2$ fold change in gene expression changes caused by *csr-1* (*tm892*) / N2 wildtype (WT) compared to RNAi knockdown of different Integrator complex subunit genes / EV that function within the catalytic or holder class. Pearson's correlation (R) value for each gene knockdown compared to *csr-1(tm892)* is shown below. N = 2133 and N = 3576 of genes up or down-regulated by >2-fold in *csr-1(tm892)* relative to wildtype are plotted for the Singh et al dataset [30] and Claycomb et al dataset [20] respectively. **f)** Linear regression analysis of genes with >2-fold change in *csr-1* (RNAi) compared to the expression of the corresponding genes in *ints-4* (RNAi) obtained from two independent studies [11,18]. N = 1353 genes plotted. For **d** and **f**, statistical significance was determined by the F-test.
(TIF)

**S3 Fig. Auxin degradation of INTS-4 on U4 snRNA processing.** Effects of ETOH or 1 mM auxin exposure beginning at the L1 stage on the levels of misprocessed and total U4 snRNA in N2 wildtype (WT) or INTS-4::degron strain. All bar graphs indicate mean ± standard error; ***<0.001 as determined by two-way ANOVA.
(TIF)

**S4 Fig. Requirement of *csr-1* for expression of 22G-RNAs targeting the Integrator. a)**
Enrichment of antisense 22G-RNAs complementary to snRNA transcripts bound to CSR-1a
or CSR-1b as determined by small RNA-sequencing data (N = 2 biological replicates) obtained
from [31]. **b)** Expression of 22G-RNAs targeting subunits of the Integrator complex in *csr-1*
*(tm892)* mutants relative to N2 wildtype (WT) as determined by total sRNA-sequencing. **c)**
mRNA expression level of Integrator subunit genes in *csr-1* RNAi or *csr-1(tm892)* mutant rela-
tive to control as determined via mRNA-sequencing and microarray using data from [20,30].
Linear regression analysis on $log_2$ fold change of Integrator subunit genes between CSR-1 KO
*(tm892)* and CSR-1 (ADH) in **d)** mRNA-seq, **e)** GRO-seq, and **f)** Ribo-seq. P<0.001 as deter-
mined by the F-test. N = 12 Integrator subunit genes plotted for **d-f. g)** Global expression of
mature piRNA levels in CSR-1 KO mutant / N2 wildtype. Data from **b-g** are obtained and ana-
lyzed from [30] with N = 2 samples per biological replicate.
(TIF)

**S5 Fig. INTS-4::mKate2 expression. a)** Representative fluorescent micrograph showing
expression of endogenous INTS-4::mKate2 in an L4 worm at the anterior region, hypodermis,
and germline after treatment with ETOH or 1 mM auxin. **b)** Representative fluorescent micro-
graph and CTCF quantification of INTS-4::mKate2 in the distal tip of dissected wildtype and
*csr-1(tm892)* mutant germline. Two independent imaging trials were performed with 612
nuclei quantified for wildtype and 478 nuclei quantified for *csr-1(tm892)*. The bar graph indi-
cates mean ± standard error, ***P<0.05 as determined by student's t-test.
(TIF)

**S1 Table. Genome-wide RNAi screen hit list.**
(XLSX)

**S2 Table. Primers used in this study.**
(XLSX)

**S3 Table. Lifespan statistics.**
(XLSX)

**S4 Table. Source data for figures.**
(XLSX)

# Acknowledgments

Some strains were provided by the *Caenorhabditis* Genetic Centre (University of Minnesota,
Minneapolis, MN) which is supported by the NIH Office of Research Infrastructure Program
(P40 OD010440). We thank Dr. Carolyn M. Phillips (University of Southern California) for
sharing the USC1258 *csr-1a(cmp135)* worm strain and Dr. Carlos Carvalho (University of Sas-
katchewan) for assistance with the DeltaVision system. BMW was supported by a USask
Devolved Scholarship.

# Author Contributions

**Conceptualization:** Cheng-Wei Wu.

**Data curation:** Brandon M. Waddell, Cheng-Wei Wu.

**Formal analysis:** Brandon M. Waddell, Cheng-Wei Wu.

**Funding acquisition:** Cheng-Wei Wu.

**Investigation:** Brandon M. Waddell, Cheng-Wei Wu.

**Methodology:** Brandon M. Waddell, Cheng-Wei Wu.

**Project administration:** Cheng-Wei Wu.

**Resources:** Cheng-Wei Wu.

**Supervision:** Cheng-Wei Wu.

**Validation:** Cheng-Wei Wu.

**Writing – original draft:** Brandon M. Waddell, Cheng-Wei Wu.

**Writing – review & editing:** Brandon M. Waddell, Cheng-Wei Wu.

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
