## [Decision Letter · Decision Letter 0]

18 Dec 2023

Dear Dr Wu,

Thank you very much for submitting your Research Article entitled 'A role for the C. elegans Argonaute protein CSR-1b isoform in small nuclear RNA 3’ processing' to PLOS Genetics.

The manuscript was fully evaluated at the editorial level and by independent peer reviewers. The reviewers appreciated the attention to an important problem, but raised some substantial concerns about the current manuscript. Based on the reviews, we will not be able to accept this version of the manuscript, but we would be willing to review a much-revised version. We cannot, of course, promise publication at that time.

If you decide to revise the manuscript for further consideration at PLOS Genetics, please aim to resubmit within the next 60 days, unless it will take extra time to address the concerns of the reviewers, in which case we would appreciate an expected resubmission date by email to plosgenetics@plos.org.

We are sorry that we cannot be more positive about your manuscript at this stage. Please do not hesitate to contact us if you have any concerns or questions.

Yours sincerely,

John Isaac Murray

Guest Editor

PLOS Genetics

Gregory P. Copenhaver

Editor-in-Chief

PLOS Genetics

Reviewer's Responses to Questions

**Comments to the Authors:**

Reviewer #1: In this manuscript the authors investigate factors that lead to misprocessing (specifically mistermination) of snRNA in C. elegans. They design a clever genetic screen for this, which produced both expected hits and many interesting hits not known to be involved in termination. Particularly interesting, they discover that the Argonaute protein csr-1 is important for correct termination of snRNA. This would be exciting and of major importance if csr-1 actually acted directly in snRNA maturation. However, the authors’ model is that at least part of the effect seems to be due to reduced expression of subunits of the integrator complex due to the fact that csr-1 normally supports expression of some target genes (though their data suggests this is not the only pathway: see below). This mechanism is less interesting- it’s not surprising that a gene that is required for expression of protein coding genes would have secondary effects due to loss of expression. Nevertheless, as a “factoid” the connection of csr-1 to this process, even if indirect, is interesting because it might provide some further insights into how csr-1 mutant phenotypes manifest in development. However, as detailed below, the authors make insufficient attempt to put their observations in the context of what is known already about csr-1. A considerable amount of analysis work would be required to address this, and probably additional high throughput sequencing. Without this I’m not sure it would be suitable for PLoS Genetics.

Major points

1. The authors’ model is that loss of the csr-1b protein results in loss of expression of integrator subunits. A lot is known and has been published about the gene expression changes in csr-1 mutants and knockdowns but the authors use their own RNAseq analysis and make no attempt to integrate this with other studies. It is imperative that the authors compare their results with other datasets from the Mello, Cecere and Claycomb labs to see how the expression of the integrator complex is affected (or not) in other analyses. If previous experiments have not revealed gene expression changes in this set of genes above what would be expected by chance, then this will significantly challenge the authors’ model because their results may be false positives.

2. Related to this, the authors do not make any attempt to correct their data for potential differences in the developmental age of their samples. This is really important because csr-1 RNAi is likely to affect development, causing many genes to vary without them being direct targets of csr-1. Straightforward tools to assess developmental age exist- see for example https://github.com/LBMC/RAPToR. These should be applied and any age differences regressed out to see if the results still stand.

3. A potential issue with the authors’ model that they should consider is how it relates to piRNAs. Csr-1 is supposed to work by preventing aberrant targeting of piRNAs to protein-coding genes. However, piRNA biogenesis, as the authors themselves cite, is promoted by integrator activity. Thus the authors model would predict that primary piRNAs would be depleted in csr-1 mutants. It’s possible that this could occur whilst still having increased piRNA TARGETING as the loss of integrator does not completely block piRNA production and remaining piRNAs could still mis-target. Or alternatively loss of csr-1 leads to increased piRNAs initially but then decreased piRNAs after loss of integrator expression. Regardless, it is clear that the authors need to examine what happens to piRNAs in csr-1 small RNA sequencing. The authors need to

i) Perform small non-coding RNA sequencing under their conditions and examine piRNAs (global levels) and 22G-RNAs targeted to integrator subunits to test if, indeed, csr-1 depletion leads to 22G-RNAs against integrator subunits being induced, and what consequence knockdown of csr-1 has to piRNAs as well as piRNA precursors (do they get misterminated, as expected from the authors’ model?)

ii) Check in the many small RNA sequencing datasets from integrator mutants to see if piRNAs are affected at all. I don’t think anyone has reported this before but they may not have looked, particularly at piRNA precursor termination defects.

4. The authors’ model of an indirect role for csr-1 function in snRNA termination is logical but as they themselves report is not sufficient to explain what csr-1 is doing because csr-1 and integrator are not fully epistatic*. The authors then need to explain what else csr-1 might be doing in this pathway. Is it acting directly on the termination as well? This would be very exciting although difficult to imagine how it might get targeted there. Are there any 22G-RNAs that are csr-1 dependent that are antisense to snRNAs?? This would be a way that snRNAs could be targeted by csr-1. Otherwise, what else is happening? Whilst it is an open ended question I really think that this is an important issue for the authors to try and provide some information about.

*Although… I think that the authors’ epistasis experiment might be confounded by the fact that they do not use complete null mutants for integrator or csr-1, which makes epistasis difficult to interpret. A better test of their model would be to try to restore integrator levels to csr-1 mutants by complementing with transgenes, though I guess that if the entire complex is affected transcriptionally this would be hard (If all subunits are limiting). If I’ve got this correct, the authors may wish to tone down their interpretation of the epistasis experiments by introducing this point as a caveat.

Minor point

Csr-1b: I’m not sure that the authors have definitively shown that this isoform is required because their experiments formally only demonstrate that csr1-a is not responsible. If they cannot specifically knock out 1b whilst preserving 1a then they need to tone down their language here to make it clear exactly what their data shows.

Reviewer #2: Review of manuscript titled “A role for the C. elegans Argonaute protein 1 CSR-1b isoform in small nuclear RNA 3’ processing” by Waddell and Wu.

In the manuscript, the authors use a genome-wide RNAi screen to investigate the regulators of the Integrator complex involved in the maturation of snRNA transcripts. They developed an elegant reporter by fusing GFP downstream snRNA transcript, which will express only in case of misprocessing of snRNA transcript. Among multiple regulators identified by screen, authors focused on Argonaute CSR-1 smaller isoform and showed RNAi of csr-1 or loss of catalytic activity resulted in snRNA misprocessing. They further show that the misprocessing of snRNA due to csr-1 knockdown is additive to the loss of integrator function. They further analysed a previously published dataset and show loss of csr-1 results in decreased translation efficiency of components of integrator complex.

Overall, the study is well done and will be important for the field.

I have a few minor comments.

1. Lines 82-85: In the introduction, the authors talk about the role of csr-1 in protection. They should also discuss that, in addition to its protective function, csr-1 has now been shown also to cleave mRNA.

2. In figure 3, the authors show the presence of an increased number of alternatively spliced transcripts. Can they comment on how their levels are compared to wt?

3. Figure 5 legend, 5a- authors have written Chip-seq data, I presume it is a typo mistake as they are showing 22G-RNAs, so it should be small RNA-seq. Please check.

4. Though authors have mentioned N in their methods, it will be useful to mention n=? for reader experience in all the figures with bar graph, number of transcripts in scatter plots and heat maps in fig 3, either as part of the figure or in the legend. Also, showing individual data points in the bar graphs wherever feasible is better.

5. To show the impact of loss csr-1 on integrator complex, authors have analysed previously published data for RNA-seq, GRO-seq and Ribo-seq for csr-1 KO, where they see reduced translation of integrator complex. Considering, they also see that rescue by catalytic inactive csr-1 also doesn’t rescue misprocessing, it will be interesting to see if the same impact of integrator transcripts is also seen CSR-1 catalytic mutant. They can analyse the data for csr-1SIN from the same publication as that for KO.

6. Finally, in the genome wide screen, authors identify in addition to csr-1 many other possible regulators which broadly categorising seems to be not only csr-1 but quite a few other members of RNAi pathway, nuclear pore proteins, ribosomal proteins among others. In the discussion, authors do talk about nuclear pore proteins, but it will be worth discussing other regulators, especially those that belong to RNAi pathways like mut-16, dcr-1, alg-4, rde-4 identified in the screen. It is possible that regulation by small RNAs might be more complex and extend beyond just csr-1.

Reviewer #3: In this manuscript, Waddell and Wu investigate the regulation of snRNA 3’ processing by the essential C. elegans Argonaute CSR-1. The authors use a clever GFP-based reporter for snRNA processing to observe defects in snRNA 3’-end termination. Using this system, they conduct a genome-wide RNAi screen and identify csr-1 among the 47 candidate genes. They show clear snRNA misprocessing in csr-1 mutant conditions, which is specific to the csr-1b isoform and requires CSR-1 catalytic slicer activity. The authors further hypothesize that components of the Integrator complex are disrupted in csr-1 mutants. To explore this hypothesis, they compare transcriptomic changes between csr-1 RNAi and RNAi targeting ints-4, an Integrator subunit. The comparison shows dysregulation at the same snRNA loci and similar overall transcriptomic profiles. To examine the genetic interaction between csr-1 and ints-4, the authors degrade INTS-4 and simultaneously knockdown csr-1 using RNAi. snRNA misprocessing defects observed upon INTS-4 degradation are exacerbated upon csr-1 RNAi, potentially due to CSR-1 regulation of multiple Integrator subunits.

To analyze the impact of CSR-1 on Integrator expression, they examine published RNA-seq, GRO-seq and Ribo-seq data. With unchanged transcription and mRNA abundance, Integrator subunits exhibit reduced ribosome occupancy by Ribo-seq. This reduced translation leads to decreased germline INTS-4 abundance, as measured by fluorescence microscopy. In summary, this study identifies a role for the CSR-1 22G RNA pathway in regulation of snRNA processing. Most of the findings are well supported by data. However, the mechanism by which this regulation occurs requires more examination. This study would be appropriate for publication in PLoS Genetics upon completion of some essential revisions. The proposed model of interaction between CSR-1 and Integrator requires more robust supporting data and a more cohesive description.

Major criticisms:

1. The authors propose a model by which CSR-1b binds 22G RNAs complementary to Integrator subunits. This complex, requiring CSR-1 slicer activity, then binds to mRNAs encoding Integrator subunits and activates translation. To support the proposed model, I suggest the authors consider a set of additional experiments.

a) To strengthen the claim that csr-1 mutants have reduced Integrator protein abundance, I suggest a more robust method such as quantitative western blots. When fluorescence microscopy is used to quantify tissue specificity of protein expression, germline tissue may be extruded and imaged to minimize background autofluorescence shown as in figure 5e.

b

---

## [Decision Letter · Decision Letter 1]

2 May 2024

Dear Dr Wu,

We are pleased to inform you that your manuscript entitled "A role for the C. elegans Argonaute protein CSR-1 in small nuclear RNA 3’ processing" has been editorially accepted for publication in PLOS Genetics. Congratulations!

Yours sincerely,

John Isaac Murray

Guest Editor

PLOS Genetics

Gregory P. Copenhaver

Section Editor

PLOS Genetics

Comments from the reviewers (if applicable):

Reviewer's Responses to Questions

**Comments to the Authors:**

Reviewer #1: I thank the authors for their considered responses to my questions in the first round of review. Whilst in an ideal world, additional sequencing would have been beneficial, I accept the authors' argument about financial constraints, and it is excellent that they have analysed existing datasets thoroughly to test their model. I should note that, whilst the authors' model to explain their observations is plausible, I'm not sure that all the data lines up behind it. In particular, it is unusual for a gene to be regulated by CSR-1 without any effect on transcript levels, as seen in the new analyses, so it is still possible that CSR-1 only indirectly regulates the integrator subunits. Nevertheless, the observation is interesting and hopefully will prompt more investigation into this potentially important effect linked to CSR-1 deficiency. I would urge the authors to make public the peer review alongside their manuscript as I think this will be of benefit to readers in evaluating the work.

Reviewer #2: The authors have done a nice job addressing the reviewers’ comments. They have included a new analysis of CSR-1 catalytic mutant from Singh et al to support requirement of catalytic activity on the integrator transcript. They have also commented on levels of alternate spliced RNA species along with improved discussion on other proteins identified in their screen in response to my comments. In response to other reviewers as well, authors have included new analysis and figures which improves the manuscript. I am satisfied and believe that the manuscript is now ready for publication.

**Have all data underlying the figures and results presented in the manuscript been provided?**

Reviewer #1: Yes

Reviewer #2: Yes

PLOS authors have the option to publish the peer review history of their article (what does this mean?). If published, this will include your full peer review and any attached files.

Reviewer #1: No

Reviewer #2: No

**Data Deposition**

http://datadryad.org/submit?journalID=pgenetics&manu=PGENETICS-D-23-01143R1

**Press Queries**

---

## [Editor Report · Acceptance letter]

8 May 2024

PGENETICS-D-23-01143R1 

A role for the C. elegans Argonaute protein CSR-1 in small nuclear RNA 3’ processing 

Dear Dr Wu, 

We are pleased to inform you that your manuscript entitled "A role for the C. elegans Argonaute protein CSR-1 in small nuclear RNA 3’ processing" has been formally accepted for publication in PLOS Genetics! Your manuscript is now with our production department and you will be notified of the publication date in due course.

With kind regards,

Anita Estes

PLOS Genetics

On behalf of:
